# Inhaled Nitric Oxide for Clinical Management of COVID-19: A Systematic Review and Meta-Analysis

**DOI:** 10.3390/ijerph191912803

**Published:** 2022-10-06

**Authors:** Jaber S. Alqahtani, Abdulelah M. Aldhahir, Shouq S. Al Ghamdi, Salma AlBahrani, Ibrahim A. AlDraiwiesh, Abdullah A. Alqarni, Kamaluddin Latief, Reynie Purnama Raya, Tope Oyelade

**Affiliations:** 1Department of Respiratory Care, Prince Sultan Military College of Health Sciences, Dammam 34313, Saudi Arabia; 2Respiratory Therapy Department, Faculty of Applied Medical Sciences, Jazan University, Jazan 45142, Saudi Arabia; 3Anesthesia Technology Department, Prince Sultan Military College of Health Sciences, Dammam 34313, Saudi Arabia; 4Department of Internal Medicine, King Fahad Military Medical Complex, Dhahran 31932, Saudi Arabia; 5Department of Respiratory Therapy, Faculty of Medical Rehabilitation Sciences, King Abdulaziz University, Jeddah 22254, Saudi Arabia; 6Global Health and Health Security Department, College of Public Health, Taipei Medical University, Taipei 11031, Taiwan; 7Centre for Family Welfare, Faculty of Public Health, University of Indonesia, Depok 16424, Indonesia; 8Institute for Global Health, Faculty of Population Health Sciences, University College London, London NW3 2PF, UK; 9Faculty of Science, Universitas ‘Aisyiyah Bandung, Bandung 40264, Indonesia; 10Institute for Liver and Digestive Health, Division of Medicine, University College London, London NW3 2PF, UK

**Keywords:** inhaled nitric oxide (iNO), acute respiratory distress syndrome (ARDS), COVID-19, hypoxemia, meta-analysis

## Abstract

Background: Severe COVID-19 is associated with hypoxemia and acute respiratory distress syndrome (ARDS), which may predispose multiorgan failure and death. Inhaled nitric oxide (iNO) is a clinical vasodilator used in the management of acute respiratory distress syndrome (ARDS). This study evaluated the response rate to iNO in patients with COVID-19-ARDS. Method: We searched Medline and Embase databases in May 2022, and data on the use of iNO in the treatment of ARDS in COVID-19 patients were synthesized from studies that satisfied predefined inclusion criteria. A systematic synthesis of data was performed followed by meta-analysis. We performed the funnel plot and leave-one-out sensitivity test on the included studies to assess publication bias and possible exaggerated effect size. We compared the effect size of the studies from the Unites States with those from other countries and performed meta-regression to assess the effect of age, year of publication, and concomitant vasodilator use on the effect size. Results: A total of 17 studies (including 712 COVID-19 patients) were included in this systematic review of which 8 studies (involving 265 COVID-19 patients) were subjected to meta-analysis. The overall response rate was 66% (95% CI, 47–84%) with significantly high between-studies heterogeneity (I2 = 94%, *p* < 0.001). The funnel plot showed publication bias, although the sensitivity test using leave-one-out analysis showed that removing any of the study does not remove the significance of the result. The response rate was higher in the Unites States, and meta-regression showed that age, year of publication, and use of concomitant vasodilators did not influence the response rate to iNO. Conclusion: iNO therapy is valuable in the treatment of hypoxemia in COVID-19 patients and may improve systemic oxygenation in patients with COVID-19-ARDS. Future studies should investigate the mechanism of the activity of iNO in COVID-19 patients to provide insight into the unexplored potential of iNO in general ARDS.

## 1. Introduction

Coronavirus disease (COVID-19), as defined by the World Health Organization (WHO), is an infectious illness caused by the severe acute respiratory syndrome coronavirus 2 (SARS-CoV-2) [1]. The main organ targeted by the coronavirus infection is the lung. The median time between the onset of symptoms and development of dyspnea, hospitalization, and acute respiratory distress syndrome (ARDS) is five days [2]. Moreover, patients with ARDS who are transferred to critical care may quickly deteriorate due to sepsis and consequent multiple-organ failure [2]. Indeed, respiratory failure due to severe acute hypoxemia is a hallmark of severe coronavirus disease and a potential target of clinical management and treatments. There is a huge burden caused by COVID-19 that may lead to death or severe sequelae [3,4], and the requirement for hospitalization, intensive care admission, and mechanical ventilation have been the major drivers of the global pressure on healthcare infrastructures [5,6]. Indeed, the development of COVID-19 vaccines presented a unique opportunity out of the various quagmires of the COVID-19 pandemic. However, the limited availability of vaccines and hesitancy in various regions of the world [7], as well as the lack of clarity on the efficacy in certain subpopulations [8,9], limit the effectiveness of the vaccines.

Nitric oxide (NO) is a systemic vasodilator endogenously produced in the endothelium [10]. The significance of inhaled NO (iNO) as a therapeutic agent is due to its ability to induce bronchodilation improving oxygen delivery to the alveoli [10]. A study conducted by Lotz et al. found that iNO administration may reduce respiratory deterioration in COVID-19 patients [11]. Another study also indicated that breathing NO improves the ventilation perfusion by lowering pulmonary arterial pressure and enhancing arterial oxygenation in patients with severe ARDS [12]. The current guidelines for the management of critically ill adults with COVID-19 advise against the regular use of inhaled nitric oxide since there are insufficient data to support its efficacy [13]. However, in light of the improvement in oxygenation, it would be fair to consider iNO in COVID-19 as a “rescue” treatment if it were accessible, but the use of iNO should be tapered down if there is a lack of improvement in oxygenation [13].

Although various studies have previously reported significant response to iNO in patients with hypoxemia or ARDS due to COVID-19 and early systematic reviews have been performed [14], there is no meta-analysis evaluating the response rate in terms of the proportion of patients who show improved systemic oxygenation following treatment. Thus, this study aimed to provide an updated systematic review of literature combined with meta-analysis of available data to clarify the clinical value of iNO in COVID-19 patients. The main research question we aimed to answer is whether iNO has significant benefit compared with standard treatment (without iNO) in the management of COVID-19-ARDS.

## 2. Methods

This systematic review was performed based on the Preferred Reporting in Systematic Reviews and Meta-Analyses (PRISMA) guidelines [15], and the protocol was prospectively registered on the PROSPERO database (CRD42022338492). Medline and Embase databases were searched in May 2022 using relevant medical subject heading (MeSH) terms systematically combined in a comprehensive search strategy including the details of searched databases, searched MeSH terms, and Boolean operators used (Appendix A). Retrieved studies were independently uploaded to EndNote 20 software (Bld 14672, Clarivate, Philadelphia, PA, USA) for duplicate removal. Duplicates-free studies were then exported for title, abstract, and full-text screening after uploading unto Rayyan software (http://rayyan.qcri.org, accessed on 22 August 2022) [16].

### 2.1. Inclusion and Exclusion Criteria

Inclusion eligibility include studies that reported the use of inhaled nitric oxide (iNO) either alone or with concomitant vasodilators as an intervention in patients with COVID-19 irrespective of treatment settings (ICU or general ward) or severity. Studies that used iNO as a potential preventative measure against SARS-CoV-2 infection, case reports, studies not published in the English language, and studies that are not available as open access or in an institutional database were excluded.

### 2.2. Data Screening and Synthesis

Screening of the titles, abstracts, and full texts was performed by two authors (TO and JSA) with resolution of conflict resolved via online meetings. Once the studies were collectively agreed upon by the screening group, the extraction of relevant data was carried out based on a predesigned narrative table. A meta-analysis was performed to pool the reported response rates to iNO in COVID-19 patients using a random- or fixed-effect model based on the between-studies heterogeneity measured by the I^2^ statistic. The response rate is defined as the proportion of patients who responded to iNO administration in terms of improved systemic oxygenation as defined by the authors. The "metaprop" algorithm on Stata/SE 16 was used, and the result was displayed as a forest plot. Response to iNO was defined as improvement (≥20%) in oxygenation following administration and was characterized by patients’ systemic oxygen saturation measured by the ratio of arterial oxygen partial pressure (PaO2 in mmHg) to fractional inspired oxygen (FiO2) expressed as a fraction (P/F ratio), and SpO2, among others. Small study effect was visually assessed by computing a funnel plot. The response rate is computed as the proportion of patients treated with iNO who were classified as “responders” after follow-up, i.e., response ratio = number of responders/treatment population, while response rate = response ratio × 100.

Sensitivity test was also performed to assess whether any of the analyzed studies disproportionately influenced the output using the "leave-one-out" algorithm. Furthermore, we performed meta-regression based on the DerSimonian–Laird (random-effect) model on the year of publication, mean age of populations that were studied, and administration of concurrent vasodilators to understand the effect on the pooled proportion that these possible confounders may have. In terms of locations of studies, we also performed further analysis comparing the response rate to iNO reported in the United States of America with other countries.

### 2.3. Quality Assessment

To assess the quality of studies included in this systematic review, a modified version of the Newcastle-Ottawa Scale (NOS) [17] was used. The scale involved 5 questions within 3 specific assessment domains covering the techniques used in patients’ "selection" in terms of sample size adequacy (≥10 patients where 10% response rate could be easily deduced) and whether the diagnosis of COVID-19-ARDS was based on the World Health Organization (WHO) or Berlin guideline [18,19], i.e., PaO2/FiO2 < 150 mmHg; "comparability” in terms of whether findings were controlled for contributory factors such age, sex, or use of concomitant therapies; and thirdly, how "outcomes" were assessed and whether patients were adequately followed up. Each question was scored with a single star if satisfied, and studies scoring ≥3 stars are classified as having low risk of bias.

## 3. Results

Search of databases generated a total of 512 studies, which included 157 duplicates. Title and abstract screening of the remaining 355 studies was performed resulting in further exclusion of 315 studies to give a total of 40 potentially suitable studies based on the inclusion criteria. The 40 studies were sorted for full-text retrieval with 2 not retrievable. A final 38 studies were then subjected to full-text screening resulting in the exclusion of 21 studies to give a total of 17 studies that satisfied the inclusion and exclusion criteria (Figure 1). The NOS assessment result showed that 14/17 (82%) of the included studies have low risk of bias (Appendix A).

### 3.1. Summary of Included Studies

Table 1 presents the features of the included studies. In summary, the 17 included studies involved 712 confirmed cases of COVID-19 of which 568 (80%) were administered with iNO alone or in combination with other vasodilators. The median (range) samples size of the included studies is 34 (10–122) patients with most studies carried out in the USA (7/17; 41%), France (4/17; 24%), and Italy (4/17; 24%). Expectedly, most study setting was the ICU (13/17; 77%) with majority of the studies designed as prospective (7/17; 41%) followed by retrospective (6/17; 35%).

### 3.2. iNO Administration

The dosage of iNO in patients with COVID-19 ranged between 9 and 160 ppm over a duration between 30 min and 5 days. The most popular mode of delivery of iNO was via mechanical ventilators (71%) in intubated patients suffering from ARDS due to exacerbation of COVID-19. In six of the included studies, iNO was administered in combination with other vasodilators such as almitrine [3,5,11], inhaled epoprostenol [7,12], iloprost [13], and ACE inhibitor or angiotensin receptor blockers [3].

### 3.3. Response to iNO

Eight (47%) of the included studies reported response to iNO in COVID-19 patients suffering from ARDS [1,2,5,6,8,14,16,17]. The eight studies included 265 COVID-19 patients administered with iNO with 166 (63%) patients reported to be responders. Response was described as improvement in oxygenation as measured by the P/F ratio, PaO2, and SpO2, among others, following iNO administration. The pooled response rate was 0.66 (95% CI: 0.47–0.84; Figure 2) with a significantly high between-study heterogeneity (heterogeneity X^2^ = 118.91, I^2^ = 94.11%, *p* < 0.001).

### 3.4. Publication Bias

The funnel plot shows the possibility of small study effect or publication bias (Figure 3), which resulted in the computation of the sensitivity test to assess whether any of the included studies overestimates the effect size (response rate). The leave-one-out sensitivity test shows that excluding any of the studies does not significantly reduce the pooled response rate, with the response rate still above 60% following consecutive removal of individual study and re-estimation of the effect size (Figure 4). This shows that none of the included studies resulted in the overestimation of the effect size.

### 3.5. Effect of Cofounders

While all eight of the meta-analyzed studies reported the year of publication and whether concomitant vasodilators were administered, only seven of these studies reported the mean or median age of the population. Where media and interquartile range of population are reported, we converted them to mean and standard deviation based on [18]. The results of the meta-regression analysis based on the DerSimonian–Laird random-effect method showed that neither the mean age of populations reported in studies (t = −1.28; *p* = 0.256), year of publication (t = 1.95; *p* = 0.099), nor the administration of concomitant vasodilators (t = 0.00; *p* = 0.999) has significant effect on the standard error of the effect sizes of the studies (Appendix A). In terms of the location of the studies, studies from the USA reported a higher response rate compared with all other countries (73%, z = 6.29, *p* < 0.001 vs 53%, z = 4.37, *p* < 0.001, Appendix A).

## 4. Discussion

This systematic review combined with meta-analysis highlighted the clinical value of iNO administration to improve the oxygenation status in patients with ARDS due to COVID-19. Our main outcome shows a response rate of 66% (95% CI: 47–84%) for patients in terms oxygenation levels following iNO therapy with or without concomitant vasodilators. Furthermore, the included studies show significant variability. However, the random-effect model that was used is robust to high between-studies heterogeneity.

Our findings contradict some previous studies. For instance, Tavazzi et al. [16] reported only a 25% response rate among patients who received iNO due to refractory hypoxemia induced by COVID-19. Moreover, Chandel et al. [6] conducted a retrospective study investigating the efficacy of the continuous use of iNO via high flow nasal cannula (HFNC) for COVID-19-ARDS patients and indicated a response rate of 39%. Both studies concluded that iNO does not significantly improve hypoxemia levels in patients with COVID-19. In contrast, three studies [1,8,17] observed significantly higher response rates of 92%, 97%, and 83%, respectively. This heterogeneity may be attributed to the different study designs and outcomes, variation in the severity of patients, type of the respiratory support used, and dosage and frequency of the iNO administered as well as concomitant therapies. However, our results suggest that iNO administration improves oxygenation in COVID-19-ARDS patients irrespective of the variabilities mentioned above.

The use of iNO in conjunction with pharmaceutical vasodilators, such as almitrine and prostaglandin, has shown a positive clinical value as a rescue therapy to enhance oxygen levels in patients with COVID-19. Two studies by Bagate et al. [3] and Laghlam et al. [11] reported significant improvement in the PaO_2_/FiO_2_ ratio after receiving iNO with almitrine (*p* <  0.01 and *p* = 0.005, respectively). Although both studies did not report a response rate and involved a small sample size, their findings were consistent with another similar study conducted by Caplan et al. [5], who reported a response rate of 66% and a significant improvement in the PaO_2_/FiO_2_ ratio in both responders compared and nonresponders to iNO combined with almitrine therapy (*p* < 0.0001). The same pattern was found in our study, suggesting that iNO with or without vasodilators is effective in the management of hypoxic patients with COVID-19. Thus, it could be hypothesized that the combined use of iNO and vasodilators can boost oxygen levels by increasing ventilation and perfusion, therefore enhancing ventilation/perfusion (V/Q) matching, reducing shunt, and improving arterial oxygenation. However, the mechanism by which this occurs requires further investigation. Furthermore, chronic lung diseases are associated with pulmonary microvascular dysfunction, contributing to the increased burden caused by such diseases [19,20]. The use of iNO in this population was promising, where iNO enhanced peak oxygen consumption because of decreased dyspnea and hyperventilation [21]. This is also evident in cardiac diseases, in which iNO therapy may be of benefit in such population [22].

Other clinical approaches to increase oxygenation levels in the management of COVID-19-ARDS patients have been described and include prone positioning [23,24] and recruitment maneuvers [24,25]. Although data that compare such supportive means with iNO are still limited, one of the included studies (19) showed that patients with COVID-19-ARDS who received iNO had significant improvement in PaO_2_/FiO_2_ (from median 136 (77–168) to 170 (138–213) mm Hg, *p* = 0.003), which increased even further after placing the patients in the prone position (from 145 (122–183) to 205 (150–232) mm Hg, *p* = 0.017). Another study [26] that used prone positioning in combination with a systemic, rather than pulmonary selective, vasodilator showed that PaO_2_/FiO_2_ was substantially increased (78.9 [27.0] vs 150.2 [56.2] mm Hg, *p* = 0.005) after receiving both therapies. These findings along with our results make a strong case for the integration of iNO with prone positioning as a routine care to enhance arterial oxygenation in patients with COVID-19-ARDS.

This study is limited by several factors. Firstly, due to underreporting, the total number of studies and, by extension, the number of patients included in our meta-analysis are low. Thus, as more data become available on the use of iNO in the clinical management of COVID-19-ARDS, future studies will further clarify this topic. However, the quality of the included studies as measured by the modified NOS assessment showed that the included studies are mainly of high quality. Secondly, considering the novelty of the SARS-CoV-2, high heterogeneity exists in the clinical management of COVID-19 patients within and across countries and continents, which results in significant between-studies variability. This is further supported by the stratified analysis (USA vs other countries) that showed that patients in the USA may have a higher response rate compared with those of their counterparts from other countries. However, we have combined a random-effect model with the measurement of publication bias; sensitivity test; as well as meta-regression controlling for age, years of publication, and administration of concomitant vasodilators to improve the interpretation and extrapolation value of our result.

In sum, this review involved a thorough and comprehensive search of the literature followed by a systematic analysis of data including a total of 265 patients diagnosed with COVID-19 who received iNO therapy. For the first time, we have reported the potential clinical value of using iNO for the management of COVID-19 patients, which suggests a relatively high benefit in terms of improved arterial oxygenation. As COVID-19 patients are at high and consistent risk of suffering from refractory hypoxemia, it is recommended that clinical guidelines and practices consider the integration of iNO in the routine clinical management plan, particularly those who are critically ill due to ARDS. Future studies should investigate whether the combined use of iNO and systemic vasodilators, or other clinical supportive means, such as prone positioning, has superior effect on the response rate and arterial oxygenation on COVID-19-ARDS population as more data become available.

## Figures and Tables

**Figure 1 ijerph-19-12803-f001:**
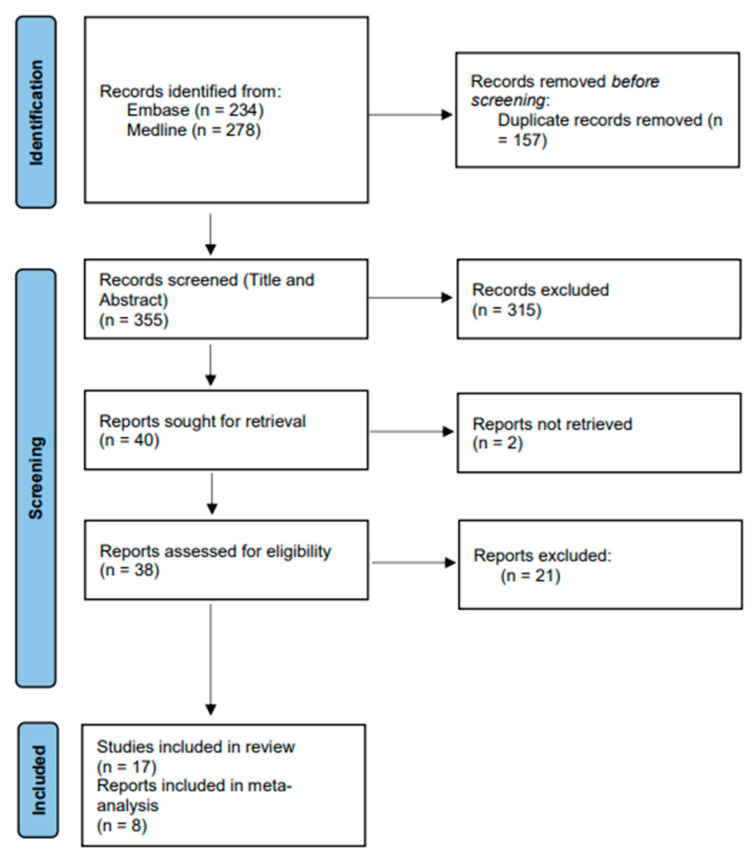
PRISMA flow chart of studies included in this systematic review.

**Figure 2 ijerph-19-12803-f002:**
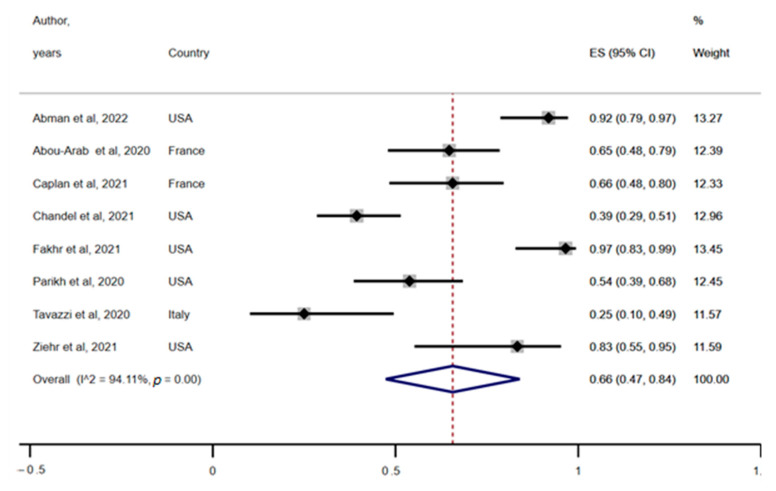
Pooled prevalence of rate of response to inhaled nitric oxide (iNO) in patients with COVID-19. Red dotted line represents overall response rate (0.66). Lateral edges of blue diamond represent extreme or limits of the 95% confidence interval (95% CI: 0.47, 0.84). ES = effect size; USA = United States of America.

**Figure 3 ijerph-19-12803-f003:**
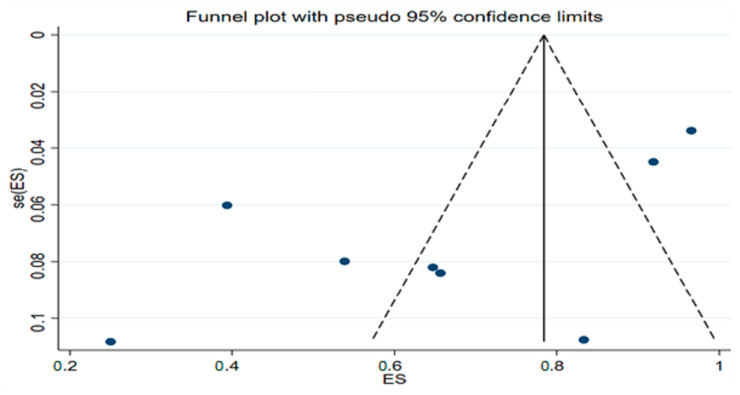
Funnel plot of studies used to compute pooled response rate to inhaled nitric oxide (iNO) in patients with COVID-19. ES = effect size; se (ES) = standard error of effect size.

**Figure 4 ijerph-19-12803-f004:**
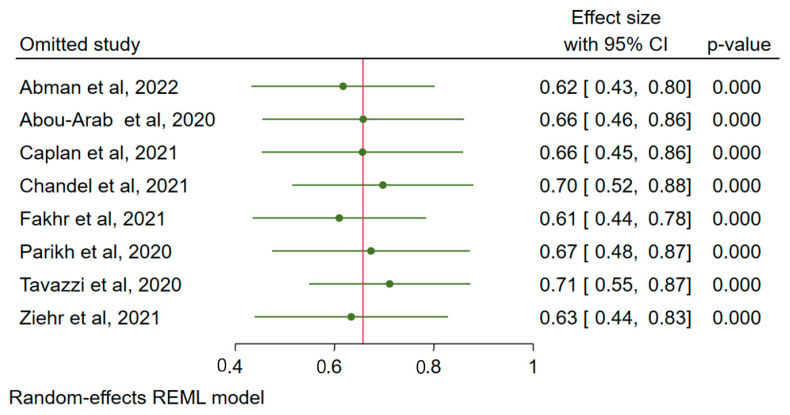
Sensitivity test (leave-one-out analysis) to detect influential studies among the reports pooled for response rate. 95% CI = 95% confidence interval.

**Table 1 ijerph-19-12803-t001:** General characteristics of studies included in systematic review.

Author, Years	Aim	Country	Study Type	Experimental Design	Settings	Sample Size (Men)	Age Mean ± SD or Median, Range	iNO Used in (No of Patients)	iNO Amount (Range)	iNO Duration	Delivery Mode	Concomitant Respiratory Stimulants	Responder	Non-Responder	Response Definition	Conclusion
Abman et al., 2022 [1]	To assess real-world iNO use and outcomes in patients with COVID-19 with mild-to-moderate ARDS	USA	Research article	Retrospective observational	General ward	37 (23)	62.0 ± 10.2	All	9–40 ppm	24 h (continuous)	Inhaled	NR	34	3	P/F increased from 136.7 (34.4) at baseline to 140.3 (53.2) at 48 h after iNO initiation	iNO was associated with improvement in the P/F ratio with no reported toxicity in hospitalized patients with COVID-19 and mild-to-moderate ARDS
Abou-Arab et al., 2020 [2]	To assess the effect of iNO administration on oxygenation in COVID-19-ARDS patients	France	Letter	Prospective observational	ICU	34 (NR)	NR	All	10 ppm	15–30 min (continuous)	Invasive mechanical ventilation	NR	22	12	PaO2/FiO2 over 20% during over 30 min following its administration of iNO	A 65% response rate to iNO was found in COVID-19 patients with severe pneumonia
Bagate et al., 2020 [3]	To assess whether inhaled iNO–almitrine combination can improve oxygenation in COVID-19-ARDS patients	France	Research article	Pilot	ICU	10 (7)	60 (52–72)	All	10 ppm	30 min (continuous)	Invasive mechanical ventilation	Almitrine and ACE inhibitors or ARB (angiotensin receptors blockers)	NR	NR	PaO2/FiO2 ratio increased from 102 (89–134) mmHg at baseline to 124 (108–146) mmHg after iNO (*p* = 0.13) and 180 (132–206) mmHg after iNO and almitrine	iNO–almitrine combination was associated with rapid and significant improvement of oxygenation in patients with severe COVID-19 ARDS
Bonizzoli et al., 2022 [4]	To assess the effect of iNO administration on cardiac function and oxygenation in COVID-19-ARDS patients	Italy	Case series	Case series	ICU	12 (8)	61.7 ± 17	All	40 ppm	24 h (continuous)	Invasive mechanical ventilation	NR	0	12	An improvement in oxygenation, as indicated by an increase in P/F ratio	iNO administration did not ameliorate oxygenation nor pulmonary hypertension in patients with severe COVID-19 ARDS
Caplan et al., 2021 [5]	To assess the effect of almitrine on arterial oxygenation in COVID-19-ARDS patients	France	Research article	Retrospective observational	ICU	32 (25)	63 (52–69)	All	10 ppm	NR	Inhaled	Almitrine	21	11	An improvement in oxygenation, as indicated by an increase in P/F ratio	Almitrine infusion improved oxygenation in severe COVID-19 ARDS patients syndrome without adverse effects
Chandel et al., 2021 [6]	To assess the effect of continuous iNO via high-flow nasal cannula (HFNC) in COVID-19-ARDS patients	USA	Research article	Retrospective observational	ICU	66 (45)	57 ± 13	All	20–40 ppm	88 h	High-flow nasal cannula	NR	26	29	Reduced need for mechanical ventilation or extension in hospital stay	iNO delivered via HFNC did not reduce oxygen requirements in most patients with COVID-19-ARDS or improve clinical outcomes
DeGrado et al., 2020 [7]	To evaluate safety and efficacy of inhaled epoprostenol and nitric oxide in patients with COVID-19-related refractory hypoxemia	USA	Research article	Retrospective observational	ICU	38 (24)	61 ± 12	11 (29)	29.1 ± 18.7 ppm	50.2 ± 31.3 h	Invasive mechanical ventilation	Inhaled epoprostenol	NR	NR	Significant change in oxygenation metrics such as P/F, PaO2, or SpO2	Inhaled epoprostenol and iNO in patients with refractory hypoxemia secondary to coronavirus disease 2019 not associated with significant change in oxygenation metrics
Fakhr et al., 2021 [8]	To assess the feasibility and effect of high-dose iNO in spontaneously breathing, non-intubated COVID-19 patients	USA	Research article	Randomized interventional	General ward	29 (16)	50 (41–60)	All	160 ppm	30 min (twice daily)	Face mask	NR	28	1	An improvement in respiratory rate of tachypneic patient’s oxygenation, as indicated by an increase in P/F ratio. Reduced need for intubation and mechanical ventilation	Administration of iNO improved the respiratory rate of tachypneic patients and systemic oxygenation of hypoxemic patients
Ferrari et al., 2020 [9]	To assess the response to iNO in mechanically ventilated COVID-19 patients	Italy	Research article	Interventional	ICU	10 (NR)	55 ± 9	All	20 ppm	30 min	Invasive mechanical ventilation	NR	NR	NR	Significant change in oxygenation metrics such as P/F, PaO2, or SpO2	iNO administration did not improve oxygenation in patients with severe hypoxemia due to COVID-19
Herranz et al., 2021 [10]	To assess the role of iNO in mechanically ventilated COVID-19-ARDS patients	Brazil	Letter	Retrospective cross-sectional	ICU	34 (24)	Median (60yrs)	12 (35)	20–20 ppm	For up to 5 days	Invasive mechanical ventilation	NR	NR	NR	Significant change in oxygenation metrics such as P/F, PaO2, or SpO2	iNO improved oxygenation (measured by PaO2/FiO2 ratio) in critically ill COVID-19 patients who are mechanically ventilated
Laghlam et al., 2021 [11]	To assess the effect of iNO and almitrine on oxygenation in COVID-19-ARDS patients	France	Brief research report	Prospective observational	ICU	12 (9)	71.8 ± 8.7	All	10 ppm	30 min	Invasive mechanical ventilation	Almitrine	NR	NR	Significant change in oxygenation metrics such as P/F, PaO2, or SpO2	Concomitant administration of iNO and infused almitrine shortly increased oxygenation in patients with COVID-19-related ARDS
Lubinsky et al., 2022 [12]	To assess the effect of iNO and inhaled epoprostenol (iEPO) on gas exchange in mechanically ventilated COVID-19-ARDS patients.	USA	Research article	Retrospective observational	ICU	84 (63)	NR	69 (49)	10–40 ppm	106 h (median)	Invasive mechanical ventilation	Inhaled epoprostenol	NS	NS	Significant change in oxygenation metrics such as P/F, oxygenation index (OI) (FiO2xmean airway pressure/PaO2), or CO2 elimination (ventilatory ratio (VR))	Inhaled pulmonary vasodilators not associated with significant improvement in oxygenation in mechanically ventilated COVID-19 patients
Matthews et al., 2022 [13]	To assess the response to iNO or prostaglandin in COVID-19-ARDS patients	UK	Research article	Prospective observational	ICU	59 (37)	60 (54–66)	48 (NR)	20–40 ppm	NR	Invasive mechanical ventilation	Iloprost	NR	NR	Significant change or improvement in oxygenation metrics such as P/F ratio	iNO and Iloprost (prostaglandin) may offer therapeutic value for ARDS-COVID-19 patients and should be investigated further
Parikh et al., 2020 [14]	To assess whether iNO therapy has any benefit for treatment of spontaneously breathing COVID-19 patients	USA	Letter	Prospective observational	General ward	39 (22)	61(NR)	All	30 ppm	2.1 days	Nasal cannula, nasal pendant with oxymizer, and nonrebreather mask	NR	21	18	Improvement in oxygenation measured by SpO2/FiO2 (SF) ratio, a surrogate for P/F ratio	iNO therapy may have a role in preventing progression of hypoxic respiratory failure in COVID-19 patients
Robba et al., 2021 [15]	To assess the effects of recruitment maneuvers (RM), prone positioning (PP), inhaled nitric oxide (iNO), and carbon dioxide removal by ECCO2R on systemic and cerebral oxygenation in mechanically ventilated COVID-19-ARDS patients	Italy	Research article	Prospective observational	ICU	22 (18)	62 [57–68.5]	9(NR)	20 ppm		Invasive mechanical ventilation	NR	NR	NR	Improvement in cerebral or systemic oxygenation PEEP, Pplat, Crs, VT, FiO2, saturation of oxygen (SpO2), pHa, PaO2, partial pressure of carbon dioxide (PaCO2), systemic (MAP, HR), and neuromonitoring parameters (TCD and NIRSderived indices)	Rescue therapy results in different effect on systemic and cerebral oxygenation in ARDS-COVID-19 patients and should be considered in choosing the right therapy
Tavazzi et al., 2020 [16]	To assess the effect of iNO administration in COVID-19 mechanically ventilated patients with refractory hypoxemia and/or right ventricular dysfunction	Italy	Letter	NR	ICU	72 (67)	66.0 [59.6–69.7]	16(NR)	20–30 ppm	15–30 min	Invasive mechanical ventilation	NR	4	12	Increase oxygenation measured by P/F ratio post administration of iNO	iNO did not improve oxygenation in COVID-19 patients with refractory hypoxemia
Ziehr et al., 2021 [17]	To understand the effect of prone position with and without iNO administration on respiratory functions in patients with COVID-19-ARDS	USA	Research article	Retrospective cohort	ICU	122 (72)	60 (51–71)	12	NR	16hr (2–36 hr)	Invasive mechanical ventilation	NR	10	2	Significant increase (>=20%) in P/F ratio	Prone positioning confers an additive benefit in oxygenation among patients treated with inhaled nitric oxide

## Data Availability

Not applicable.

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
