# Peer review of "Inhaled Nitric Oxide for Clinical Management of COVID-19: A Systematic Review and Meta-Analysis"

_ijerph, 2022, doi:10.3390/ijerph191912803_

Round 1
Reviewer 1 Report
The review is a combination of already published facts. It is a compilation of some of the literature published from 2020-2021. It does not bring out any thought-provoking scientific/ mechanistic insights on the role of inhaled nitric oxide in treating covid-19
Author Response
The review is a combination of already published facts. It is a compilation of some of the literature published from 2020-2021. It does not bring out any thought-provoking scientific/ mechanistic insights on the role of inhaled nitric oxide in treating covid-19
Response: We thank the reviewer for this comment and indeed, we agree that this is a compilation of published work regarding the use of inhaled nitric oxide in management of COVID-19 complications. The aim was to push an argument based on the published facts about the usefulness of nitric oxide (a systemic vasodilator) in the management of acute respiratory distress syndrome associated in severe COVID-19 cases and the goal is to help clinicians and practitioners understand how it has been used and the possible reported outcome of using iNO in treatment of COVID-19 ARDS. We believe this is a central tenet of systematic reviews and this is even strongly so when combined with rigorous meta-analysis for effect sizes. However, we are happy to improve on the work based on any suggestion to improve the readership and strengthen the argument. We sincerely thank you once again for taking out of your busy schedule the time to go through our manuscript.

Reviewer 2 Report
This meta-analysis aimed to provide insightful information on using inhaled nitric oxide for clinical management of COVID-19 with acute respiratory distress syndrome. PRISMA guidelines for a systematic review was followed.
You described in "3.2. iNO administration" (Lines 150 - 152): In six of the 8 included studies, iNO was administered in combination with other vasodilators, but there is no mention on how you isolated the pure iNO effect from others that were co-administrated. A major revision for clarifying such isolation in meta-analysis is advised.
Author Response
This meta-analysis aimed to provide insightful information on using inhaled nitric oxide for clinical management of COVID-19 with acute respiratory distress syndrome. PRISMA guidelines for a systematic review was followed.
You described in "3.2. iNO administration" (Lines 150 - 152): In six of the 8 included studies, iNO was administered in combination with other vasodilators, but there is no mention on how you isolated the pure iNO effect from others that were co-administrated. A major revision for clarifying such isolation in meta-analysis is advised.
Response: We thank the reviewer for this insightful comment. Indeed, it was difficult to isolate the effect of iNO since in most case it is administered alongside other therapies including antivirals and other vasodilators. However, we have performed meta-regression analysis on the administration of other vasodilators as well as mean age of populations as well as years of publication and no significant effect was found on the effect sizes reported.

Reviewer 3 Report
Authors used meta-analysis to review the inhaled nitric oxide for clinical management of COVID-19. It is a novelty paper in this field, but more problems still appear in this paper. I hope that the author would revise the manuscript properly in response to the following suggestions
1. In the abstract, “and resulted in improved systemic oxygenation in patients with COVID-19-ARDS”. this conclusion does not come from your study.
2. The authors do not specify the questions. Please address the questions in the introduction section.
3. In the method section, I don’t know how the effect size is calculated and what index is used to represent it. The standardized mean difference? Or response ratio? You need to make this very clear in this section.
4. And, how do you Screen the papers, and Which database are these papers from? which keywords and boolean operators (AND, OR, NOT) did you use for your search?
5. Please write in detail in 2.2. Data screening and synthesis. The reader needs to repeat your research process.
6. Funnel plot is not enough for Publication Bias, please regression methods (Egger’s regression) or the trim and fill method to further confirm the results
7. please add a TABLE to show the overall effect of fixed and random models.
8. please show the Q value in the heterogeneity test.
9. please examine the effect of publication year to effect size using meta-regression analysis. This allows for assessing whether the year of publication affects the ES.
10. The experimental design in these papers is not described clearly in this study. pre-test/post-test design or experimental and control groups design?
11. line124-131: I suggested this paragraph can be moved to 2.2 Data screening and synthesis. Because this is not the results of the meta-analysis
12. I suggested author could compare the difference in effect size between different countries, for example, the USA vs other countries (France, Italy, Brazil), you could set countries as the moderator.
Author Response
Authors used meta-analysis to review the inhaled nitric oxide for clinical management of COVID-19. It is a novelty paper in this field, but more problems still appear in this paper. I hope that the author would revise the manuscript properly in response to the following suggestions
- In the abstract, “and resulted in improved systemic oxygenation in patients with COVID-19-ARDS”. this conclusion does not come from your study.
Response: We thank the reviewer for this comment. We have rephrased the statement to accommodate the limits of our findings considering the limitations of the study.
- The authors do not specify the questions. Please address the questions in the introduction section.
Response: Thanks again for this comment. We have specified the research question in the closing statement of the introduction section.
- In the method section, I don’t know how the effect size is calculated and what index is used to represent it. The standardized mean difference? Or response ratio? You need to make this very clear in this section.
Response: Thanks. We have updated the method to include how the effect size was computed and the index used to present it. Specifically, we have
- And how do you Screen the papers, and Which database are these papers from? which keywords and boolean operators (AND, OR, NOT) did you use for your search?
Response: Thanks again. We have included details on the search strategies (Line 86-87) and ask that a copy that was submitted with the original manuscript be promptly provided to you. Thanks.
- Please write in detail in 2.2. Data screening and synthesis. The reader needs to repeat your research process.
Response: Thanks again for this comment. We have included more details on the data screening and synthesis.
- Funnel plot is not enough for Publication Bias, please regression methods (Egger’s regression) or the trim and fill method to further confirm the results.
Response: We very much thank the reviewer for these insightful comments. We agree that funnel plot may not give the entire picture of the bias associated with our analysis. Thus, we explore meta-regression as advised. We have included details of this further analysis as well as result (figures) as supplemental material.
- Please add a TABLE to show the overall effect of fixed and random models.
Response: Thanks for this comment. We did not perform fixed effect model because it is not robust to high between-studies heterogeneity. This is the reason why all analysis performed were down using random effect model to correct for the variability.
- Please show the Q value in the heterogeneity test.
Response: Thanks, we have included the heterogeneity X^2 in Line 165. This is the only measure of heterogeneity provided by Stata SE.
- Please examine the effect of publication year to effect size using meta-regression analysis. This allows for assessing whether the year of publication affects the ES.
Response: Thanks. We performed a meta-regression and the result and description is now provided.
- The experimental design in these papers is not described clearly in this study. pre-test/post-test design or experimental and control groups design?
Response: Thanks, we have rephrased the table 1 to clarify the experimental design.
- Line 124-131: I suggested this paragraph can be moved to 2.2 Data screening and synthesis. Because this is not the results of the meta-analysis.
Response: Thanks for this comment. We have looked into this and did a background check on the method of reporting in systematic reviews and meta-analysis and based on this we prefer to leave it as part of the result based on screening of previous systematic reviews. Thanks again.
- I suggested author could compare the difference in effect size between different countries, for example, the USA vs other countries (France, Italy, Brazil), you could set countries as the moderator.
Response: Thanks, we performed this analysis and updated the discussion to include the finding. The figure is shown in supplemental figure 5.

Round 2
Reviewer 3 Report
minor comments:
1. please add the comparison result between different countries in ABSTRACT.
2. other publication bias tests should be moved after the result of the funnel plot. You could set subheadings for the publication bias. for example, 3.4 publication bias.
3. in the results, can you provide the Q-value and P level for the comparison between different countries?
4. Can you provide the Calculation formula of effect size (response ratio) in the methods section?
Author Response
- please add the comparison result between different countries in ABSTRACT.
Response: Many thanks for this comment. We have added a detail of the comparison result to the abstract as recommended. Thanks again.
- other publication bias tests should be moved after the result of the funnel plot. You could set subheadings for the publication bias. for example, 3.4 publication bias.
Response: Thanks again for this comment. We have now created a new subheading for all results of publication bias. Thanks.
- in the results, can you provide the Q-value and P level for the comparison between different countries?
Response: Many thanks. We have included the z score and P-level for the comparison in the result section (73%, z = 6.29, p < 0.001 vs 53%, z = 4.37, p < 0.001). Accordingly, this are the tests that the countries compared have effect sizes of 0 (H0).
- Can you provide the Calculation formula of effect size (response ratio) in the methods section?
Response: We thanks the reviewer for this comment. We have included the formula for calculation of the effect size (response ratio and indeed, response rate) in the methods section for clarity. Many Thanks.
